# Interplay of structured and random interactions in complex ecosystems dynamics

Juan Giral Martínez[1]*, Matthieu Barbier[2,3,4], Silvia De Monte[1,5]

**1** Institut de Biologie de l'École Normale Supérieure, Département de Biologie, École Normale Supérieure, PSL Research University, Paris, France, **2** CIRAD, UMR PHIM, Montpellier, France, **3** PHIM Plant Health Institute, University of Montpellier, CIRAD, INRAE, Institut Agro, IRD, Montpellier, France, **4** Institut Natura e Teoria en Pirenèus, Surba, France, **5** Max Planck Institute for Evolutionary Biology, Plön, Germany

☯ These authors contributed equally to this work.
* jgiral@bio.ens.psl.eu

## Abstract

Minimal models for complex ecosystems often assume random interactions, whose statistics suffice to predict dynamical and macroecological patterns. However, ecological networks commonly possess a variety of properties, such as hierarchies or functional groups, that structure species interactions. Here, we ask how conclusions from random interaction models are altered by the presence of such community-level network structures. We consider a Lotka-Volterra model where pairwise species interactions combine structure and randomness, and study macroscopic community-level observables, abundance distributions and dynamical regimes. Randomness and structure combine in a surprisingly yet deceptively straightforward way: contributions from each component to community patterns are largely independent. Yet, their interplay has non-trivial consequences, notably out of equilibrium. We conclude that whether interaction structure matters depends on the pattern: when breaking species equivalence, static patterns of species presence and abundance predicted from random interaction models are less robust than the qualitative nature of dynamical regimes.

## Author summary

Ecological communities with many species are a challenge for theoretical ecology, since they harbour many and very diverse interactions. A common approach is to focus on aggregated quantities, such as the abundance of broad taxonomic or functional groups, thus neglecting all the diversity at finer scales. We develop a theoretical framework that combines both ecological structure and fine-grained heterogeneity, which we represent with randomness. We find that structure and randomness contribute to ecosystem properties with effects that can be largely disentangled, allowing a continuous transition between two simple effective

**Data availability statement:** The source code used to produce the results and analyses presented in this manuscript is freely available at the GitHub repository: https://github.com/juanelogiral/interplay-structure-randomness.git.

**Funding:** MB was supported by the European Union (GA#101059915 - BIOcean5D) https://research-and-innovation.ec.europa.eu. JGM was supported by the Frontiers in Research and Education graduate program https://phd.learningplanetinstitute.org/. The funders had no role in study design, data collection and analysis, decision to publish, or preparation of the manuscript.

**Competing interests:** The authors have declared that no competing interests exist.

descriptions of the community dynamics. Still, they interact in unexpected ways. For instance, we show that species heterogeneity can simplify the dynamics of functional groups, countering the idea that complexity should always destabilize large ecosystems.

## 1 Introduction

Ecological communities, from microbiomes to food webs, may appear intractably complex due to the large number of entities and processes they harbour over a wide range of scales. One particular facet of this complexity is that, besides spatial, environmental or intra-species processes, community-level patterns and dynamics stem from interactions between a vast number of different species.

The founding work of Robert May [1] proposed to represent such complexity by drawing interactions randomly in a linear model, thus focusing on predictable statistical consequences of intractably complex species relations. Inserting random interactions into classic population dynamics models, like the generalized Lotka-Volterra Equations (gLVEs), later allowed to leverage methods from the statistical physics of disordered systems [2] to describe the dynamics of systems with a large number of degrees of freedom. The dimension of the relevant parameters' space is thus drastically reduced, facilitating comparison with observational data. Only a few interactions' statistics indeed turned out to be essential for explaining patterns of ecological communities, such as abundance distributions and transitions between dynamical regimes [3–5]. Such models have proved appealing because of the simplicity and range of their predictions, and have found qualitative corroboration in recent experiments [6,7].

However, it is unclear how widely the conclusions drawn from these "disordered models" can be expected to hold. Indeed, because they treat all interactions as independent and species as statistically equivalent, they openly neglect dynamically significant ecological structure. This assumption may hold within a guild of relatively similar species, but broader ecological networks are usually conspicuously structured, and there is evidence that this structure is reflected in dynamics (e.g. highly unstable species dynamics, yet stability at the level of higher taxonomic [8,9] or functional groups [10–12], although not always more than expected from simple aggregation [13]).

Our goal here is to investigate how predictions of fully random models are altered when species interactions possess a "macroscopic", community-level structure. This objective has motivated a vast array of theoretical studies, yet it is difficult to extract a general perspective from the literature, because of the sheer diversity of possible structures and the lack of a unified methodology and set of predictions to compare. Indeed, every possible structure potentially opens up new questions, such as asking about resource use efficiency and sequence in random consumer-resource models [14,15], or trophic coherence in more complex food webs [16].

PLOS Computational Biology

Among possible questions, (linear) equilibrium stability has been studied most systematically: a range of studies have extended May's original approach, using developments in Random Matrix Theory to combine randomness with various specific structures, e.g. bipartite or trophic networks, spatial clusters, sparse interactions or complex correlations [17–19]. Departures from this linear approach have gone in very diverse directions, and while the popularization of some techniques (e.g. Dynamical Mean Field Theory, presented below) has focused efforts on a more unified subset of predictions, studies have still found disparate consequences to the addition of structure: causing stability to increase, decrease or both [20,21]; affecting coexistence positively or negatively [3,22,23]; inducing distinct dynamical phases [24,25]. We thus ask whether there could exist any broad principles in the interplay of randomness and structure, which, though perhaps not capturing the full range of these explorations, can provide baseline expectations on what features of random models are broken or conserved when interactions are partially structured [26].

We consider a dynamical model of interacting populations, and assume that their interactions derive from many ecological processes, which can be partitioned into a large set of *hidden* processes, and a small set of *structural* processes. The hidden processes are represented by drawing interactions independently at random, as they arise from many different mechanisms [6] that are fixed but unknown, and may vary from context to context (hereafter, different "realizations" of the randomness). The structural processes reflect the modeler's ecological knowledge of the community, and are tied to broad ecological functions or mechanisms that many or all species participate in. For instance, structural interactions between any two species may be shaped by their impact on and response to key public goods (e.g. resources, pH), or through trophic relations mediated by traits (e.g. body size). Interactions may also be determined by species belonging to distinct taxonomic or functional groups (e.g. 'plants', 'herbivores' and 'carnivores', 'phytoplankton' and 'zooplankton') whose total biomasses serve as macroscopic functional variables, e.g. in low-complexity models in ecology [27] . Key to predictability is a reduction in system complexity. In both fully structured and fully random models, this is achieved by coarse-graining, i.e. identifying a few macroscopic variables that account for static or dynamical properties of the community . Here, we study whether a coarse-grained description still exists as we interpolate between those two extremes, and what form it takes.

We show that the mathematical approach of disordered systems can be applied to a general model that combines randomness with a broad class of structures. The community dynamics is therefore epitomized by a few community-level variables and parameters, associated to the structuring functions and to randomness. These variables are related to each other through closed relations and thus provide a self-sufficient macroscopic description of the community. At equilibrium, they are enough to determine all other macroscopic patterns, such as species abundance distributions. Moreover, we find that randomness and structure can interfere in ways that unexpectedly impact community-level outcomes, as we discuss when the structure-driven dynamics is out-of-equilibrium.

## 2 Methods

### 2.1 Model for the dynamics of species-rich, interacting communities

We describe the dynamics of the rescaled abundances $x_i(t)$ of $S$ interacting species, labeled $i = 1, \ldots, S$, in the framework of generalized Lotka-Volterra Equations. This is the simplest non-linear model capturing the effect of ecosystem composition on individual demographic rates, which are influenced by both intra-specific competition and inter-specific interactions. In order to focus on heterogeneity in the species' interactions, we consider a dimensionless version of the equations, which is derived in the Appendix A in S1 Text from a more general formulation in terms of true population abundances.

The variation in time of the rescaled abundance of species $i$

$$\frac{dx_i}{dt} = x_i \left[ 1 - x_i + \sum_j A_{ij} x_j \right] + m. \tag{1}$$

depends on the *interaction coefficients* $A_{ij}$ that account for the impact of species $j$ on the instantaneous growth rate of species $i$. Moreover, a small *immigration rate* $m \ll 1$ (set here to $m = 10^{-8}$) is added to avoid species getting permanently excluded from the community [4,28]. Species that persist at the immigration level can therefore invade as soon as the community context becomes favorable. At any given time in the simulations, however, species whose abundance is smaller than a multiple of $m$ are counted as extinct in order for immigration not to artificially inflate species diversity.

We model the interaction matrix **A** as the superposition of a *structural* and a *random* component (schematically illustrated in Fig 1). The first encodes our ecological knowledge of the community and the second all latent interactions that are too intricate to be accurately characterized. Other potential sources of stochasticity, such as demographic or environmental fluctuations, are not included in the model. When the matrix is completely unstructured, Eq (1) reduces to the so-called disordered gLVEs, whose dynamics has been extensively characterized under different assumptions on the statistics of the random interactions [4,22,28,29]. The structural part, on the other hand, is often modeled explicitly in various ways, e.g. with trophic or size classes, or the use of metabolic substrates and the production of public goods [8,30,31], but it is reduced here to its effect on pairwise interaction coefficients. The two terms of such decomposition can generally vary in their relative importance, bridging between two extreme cases that have been previously addressed with distinct mathematical approaches.

## 2.2 Structured interactions

We define structure as resulting from the existence of a set of $n_F$ relevant community-level functions, which will be in limited number by construction. Functions $f_\lambda$ are indexed with Greek letters, $\lambda = 1, \ldots, n_F$. Every species can contribute to a community function and can be affected by it (see examples in Fig 1). This relationship is characterized by two sets of species-specific *functional traits* (as illustrated in Fig B in S1 Text). The *impact trait* $\mathcal{I}_i^{(\lambda)}$ quantifies the per-capita contribution of species $i$ to function $\lambda$. The *sensitivity trait* $\mathcal{S}_i^{(\lambda)}$ gives the impact of function $\lambda$ on the per-capita growth rate of species $i$ [32]. These can also be interpreted, along with [33], as the impact and requirement niches, respectively.

The total *magnitude of the function* $f_\lambda$ at time $t$ is for simplicity chosen as the sum of all species' contributions, resulting in the linear combination of the abundances

$$f_\lambda(t) = \overline{\mathcal{I}^{(\lambda)} x(t)} := \frac{1}{S} \sum_i \mathcal{I}_i^{(\lambda)} x_i(t). \tag{2}$$

Here and in the following, the overline defines an average over species for a fixed set of parameters (interaction coefficients). Due to the factor $S^{-1}$, each species's contribution to a given function is small in comparison to that of the rest of the community, in accordance with our focus on collective functions – which cannot be achieved by one species alone. Similarly, the total impact of functions on a species' growth rate is assumed to be the sum of the functional magnitudes, weighted by the sensitivity traits, i.e. $\sum_\lambda \mathcal{S}_i^{(\lambda)} f_\lambda(t)$.

Interactions mediated by collective functions are then entirely encapsulated in a *structural matrix*

$$\mu_{ij} = \frac{1}{S} \sum_\lambda \mathcal{S}_i^{(\lambda)} \mathcal{I}_j^{(\lambda)} \tag{3}$$

where the effect of any species on another depends on their functional traits. Conversely, any interaction matrix with collective interactions can be written in the form of a product of trait vectors via its Singular Value Decomposition, even though the functional traits thus obtained need not correspond to known ecological processes. We show in Appendix B in S1 Text that, due to the scaling choices in Eq (3), we can always restrict ourselves to $n_F \ll S$ (i.e. our structural matrix is always low-rank) with important consequences for our results.

Our approach can in principle be extended beyond linear community functions. Linearity can be relaxed either in the definition of the functions, or in their contribution to the species growth rate, thus encompassing interaction structures that

depend nonlinearly on traits, e.g. trophic niche models based on body size differences [34], as well as saturating functional responses or higher-order interactions. Some of these possibilities can be captured by straightforward modifications of our analysis, following existing work [5,35–37], while others may require a more complex treatment, e.g. [38].

Fig 1 illustrates the structural matrices derived in three examples of communities structured by functional traits. When functional or phylogenetic groups exist [39], all species of a group have an even impact on its biomass, i.e. $\mathcal{T}_i^{(\lambda)} = 1$ if the species is part of the group and 0 otherwise, and we assume that species are sensitive to the total biomass of the groups. When community functions are public goods or resources, species impact and sensitivity are based on their consumption (or production) of those compounds, and on the yield, respectively [15]. Most of our later examples will be illustrated in the more intuitive case of non-overlapping functional groups, but the generality of our analytical approach is demonstrated in Appendix I in S1 Text.

### 2.3 Combining structure and randomness

Structure being thus defined, we now turn to the unstructured part of the interactions, reflecting sources of between-species variation not described by community functions. These non-observable features are modelled as random terms [1,22,40], and are defined as a *random matrix* of entries $\sigma z_{ij}$ (Fig 1D). For simplicity, the $z_{ij}$'s are chosen as independent,

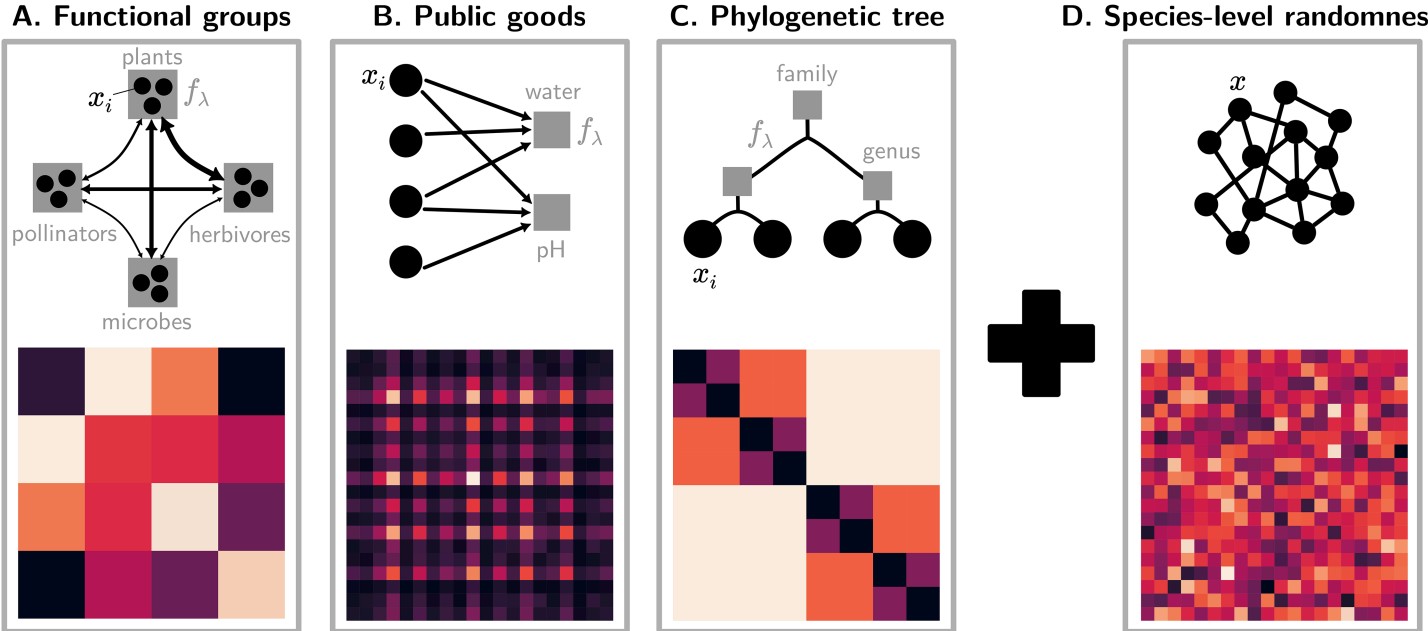

**Fig 1**. **Examples of community-level structure and species-level randomness in species interactions.** We model interactions between individual species abundances $x_i$ resulting from the superposition of structured (left panels) and random components (right panel), whose relative magnitude can be tuned. Three examples of ecosystem structures (top row) produce different species interaction matrices $A_{ij}$ (bottom row). Structure in each matrix follows Eq (3), where interactions between species (black disks) are mediated by a set of community-level processes (black arrows) involving collective functions $f_\lambda$ (gray squares) defined in Eq (2). In our three examples, species interactions depend on: **A.** the total abundance $f_\lambda$ of a small set of functional groups (e.g. plants, herbivores...), so that the block structure of the matrix reflects the group structure of the community (different colors indicate different values of interaction between pairs of groups); **B.** the availability or magnitude $f_\lambda$ of a small set of public goods (e.g. resources or environmental variables), in which case the rank of the matrix is bounded by the number of public goods, resulting in highlighted lines and columns; **C.** traits that depend on phylogenetic distance, as encoded by the fractal structure of the interactions (larger blocks correspond to higher-order taxa, and block colour to the intensity of interactions between pairs of taxa), in which case $f_\lambda$ are the total biomasses of subtrees of various depths. **D.** other unknown or "hidden" processes that affect every species independently are captured by the addition of interaction terms sampled at random from a distribution with given statistics.

identically distributed variables with zero mean and unit variance. The specific details of the probability distribution do not matter, as long as it has finite variance and doesn't depend on $S$. A generalization to $z_{ij}$ with non-identical variances is provided in Appendix G in S1 Text.

The *total interaction matrix* $A_{ij}$ is defined as the sum of the structured and of the disordered components

$$A_{ij} = \mu_{ij} + \frac{\sigma}{\sqrt{S}} z_{ij}. \tag{4}$$

The $S^{-1/2}$ scaling factor, again, ensures that single interaction terms are small compared to the aggregated effect of the whole community. Given that $n_F \ll S$ (see above and Appendix B in S1 Text), the total interaction matrix is the superposition of a low-rank and a random term.

The parameter $\sigma$ balances the relative importance of structure and randomness. Without structure, Eq (1) reduces to the classical disordered gLVEs studied in statistical physics. Accordingly, we will consider typical ecosystem outcomes, i.e. those that do not depend on the specific realization of randomness. The expectation is that, when structure is added, the statistical properties of the random term should be sufficient to characterize behaviours that are independent of the unknown species-level details. In the spirit of macroecology, moreover, such generic predictions are expected to be connected with recurrent empirical patterns .

## 3 Results

In the following Sect 3.1, we obtain an effective description of the properties of a community by applying Dynamical Mean Field Theory (DMFT, [2,4,28,41]) to Eq (1) with interactions that combine structure and randomness, in order to understand the interplay between their respective predictions. We use it in Sect 3.2 to draw conclusions on how these two components combine at equilibrium. We first derive a set of closed equations for community-level degrees of freedom, that determine both community- and species-level states, and illustrate their consequences by focusing on Species Abundance Distributions, a pattern classically studied in theoretical as well as applied ecology [42]. Finally, in Sect 3.3 we discuss how such community equilibrium can lose stability, and the respective roles of randomness and structure in out-of-equilibrium species coexistence. For the sake of readability, the main text only presents mathematical keystones, while the analytical details can be found in Appendixes D, E, F and H in S1 Text. The precise numerical protocols can be found in Appendix C in S1 Text.

### 3.1 Effective community dynamics reflects both structure and disorder

In the absence of randomness, $\sigma = 0$, the Lotka-Volterra system's behaviour is entirely determined by the structured relations between species. We can rewrite Eq (1) as

$$\frac{dx_i}{dt} = x_i \left[ 1 - x_i + \sum_\lambda s_i^{(\lambda)} f_\lambda(t) \right] + m \tag{5}$$

and thus express the dynamics in terms of the functional variables $f_\lambda$. In this case, the equilibrium abundances of species are exclusively set by the community-level variables, with multiple species (that share the same interaction coefficients) collapsing onto the same abundance (Fig 2A, left). For instance, when modeling how the total abundance of plants depends on that of herbivores, or how pH depends on water availability, one considers that all plants are the same, or that all species respond to the same environmental variable. Such strict correspondence between the microscopic and macroscopic scales also holds in the dynamics (Fig 2A, right), where equivalent species asymptotically follow the behaviour of their average abundance (thus determining macroscopic variations of the functional variables), even when these cancel

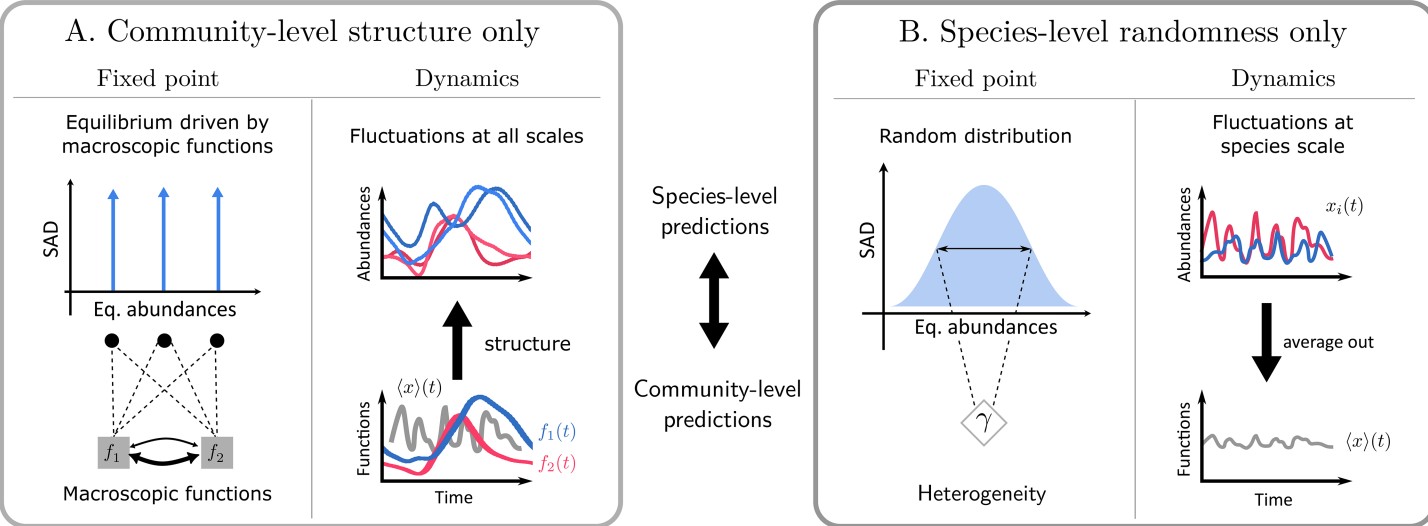

**Fig 2**. **Theoretical expectations from either structure or randomness in species interactions.** With the modeling choices in Eqs (3) and (4), both purely structured and purely random species interactions allow species-level predictions, such as Species Abundance Distributions (SAD) and stability, but also a low-dimensional effective description of the whole community. **A.** Interactions structured by a small number (compared to the number of species) of community-level degrees of freedom define classes of functionally equivalent species. Community-level descriptors are then recapitulated by the collective functions and their relations, both at equilibrium (left) and out-of-equilibrium (right, where functions in colour are, for instance, the mean $x_i(t)$ over all species of the same functional class). Functionally equivalent species keep a coherent dynamics that emerges at the macroscopic level. **B.** When interactions are randomly assigned, the static community-level patterns (abundance distribution, left) are accounted for by statistical metrics, e.g. the degree of dispersion in species interactions $\gamma$. Out of equilibrium, complex dynamics can manifest at the species level, but average out in macroscopic observables (right, note that here macroscopic functions represented in gray are averages over the whole unstructured community).

out when averaging over the whole community. Community-level static or dynamical relations yield various ecological patterns such as biomass pyramids, trophic cascades or regime shifts, see e.g. [43,44], which directly trickle down to species scale as long as species are held equivalent.

Once randomness is introduced, the connection between community-level and single-species dynamics is less straightforward (Fig 2B). DMFT provides an appropriate framework for studying aggregated properties that are shared by typical realizations of the disordered matrix [4]. This approach describes aggregated, *community-level* variables, usually called "self-averaging", that are the same for any (large) community whose fixed set of interactions is randomly sampled from a distribution with given variance. Contrary to such coarse-grained quantities (represented by $\gamma$ in Fig 2B left), *microscopic variables* such as the abundance of a single species (whose distribution is represented in Fig 2B left) generally change from one to another realization (sampling) of randomness.

DMFT can be generalized to encompass structure and disorder in the interaction matrix Eq (4). We summarize in the following the essential steps of the derivation of such Structured DMFT, and leave the details for Appendix D in S1 Text. A key feature is that, due to the large number of species, a formal equivalence can be established between the original dynamical equations with random interaction coefficients and a set of $S$ uncoupled Stochastic Differential Equations (SDE)

$$\frac{\mathrm{d}x_i}{\mathrm{d}t} = x_i\left[1 - x_i + \sum_\lambda \mathcal{S}_i^{(\lambda)} f_\lambda(t) + \sigma \zeta_i(t)\right] + m \tag{6}$$

where the dynamics of species $i$ depends, via the species-specific sensitivity traits, on the magnitude of functions $f_\lambda$ and on a Gaussian colored noise $\zeta_i$. The noise term in Eq (6) depends on the abundances of all species in the

community through the relation $\langle \zeta_i(t)\zeta_j(t') \rangle = \delta_{ij}C(t,t')$, where $C(t,t')$ is the two-time average of abundance, or *community correlator*,

$$C(t,t') = \overline{x(t)x(t')} = \overline{\langle x(t)x(t') \rangle}. \tag{7}$$

Here, brackets indicate averages over a single species in multiple realizations of the random matrix, while the overline is an average over species for a given realization, as defined in Eq (2). Owing to the law of large numbers, averages over species equal averages over species and randomness. This implies that, as for the ordinary DMFT, the correlator doesn't depend on the matrix realization.

Similarly, for a fixed value of randomness intensity $\sigma$, functional variables are self-averaging and can thus be directly computed from the abundances averaged over realizations of the randomness:

$$f_\lambda(t) = \overline{\mathcal{J}^{(\lambda)}\langle x(t) \rangle}. \tag{8}$$

Such averaging makes them independent of species-specific details represented by the random contributions $z_{ij}$, while they retain the structural features through the effect traits $\mathcal{J}_i^{(\lambda)}$. The most significant difference with prior studies is that, while DMFT in the usual setting yields a low-dimensional model, i.e. a single equation representing the distribution of possible trajectories of any species picked at random in the system, here Eq Eq (6) are still as numerous as in the initial model, because species are intrinsically different from each other due to their distinct traits $\mathcal{S}$'s and $\mathcal{J}$'s; what has been gained is that the equations are now uncoupled.

Due to the non-linear nature of Eq (1), the simplicity of the equations does not entail simplicity in the dynamics. Every species is influenced by $n_F + 1$ community-level, self-averaging quantities: the functional magnitudes Eq (8) and the correlator Eq (7). Just as the functions are sufficient to capture the effect of structure in the absence of disorder, the effects of disorder are summarized by the sole correlator and are not species-specific.

The statistical equivalence between the original dynamics and trajectories obtained from the coarse-grained equations hinges upon such community-level observables being self-consistent, that is matching the microscopic, species-level stochastic processes. Their computation can in principle be achieved numerically [28], as in the unstructured case. The algorithm however would have important computational costs, exacerbated in our case by the fact that different species have different statistics. We therefore concentrate here on exploiting analytical results that can be obtained at equilibrium, and use them to gain also insight on the onset of out-of-equilibrium regimes.

## 3.2 A closed set of community-level variables determines equilibrium patterns

We show that, when the community is at equilibrium, the previously defined coarse-grained variables obey closed equations, which in turn allow us to characterize species abundance distributions. In the next section, we will discuss the conditions for such an equilibrium to be stable.

Here, we focus on the Species Abundance Distribution (SAD), *i.e.* the fraction of species of at a given equilibrium abundance $x_i^\star = \lim_{t\to\infty} x_i(t)$. A long tradition in ecology looks for shared patterns in such distributions, and compares them with the predictions of theoretical models that typically encompass randomness but not structure [45–47] (but see [48]).

At equilibrium, the $n_F$ functional magnitudes and the correlator attain the values

$$\begin{aligned} f_\lambda^\star &= \lim_{t\to\infty} f_\lambda(t) \\ C^\star &= \lim_{t\to\infty} C(t,t). \end{aligned} \tag{9}$$

As discussed in Sect 3.1, the equilibrium abundances of individual species are not self-averaging, and therefore depend on the realization of randomness. Still, it is useful to investigate how macroscopic structure biases species

towards smaller or higher abundances. We do so by defining the *modal abundances*:

$$x_i^+ = 1 + \sum_\lambda \mathcal{S}_i^{(\lambda)} f_\lambda^\star,$$ (10)

which we will later prove to correspond to the most likely abundance at equilibrium of extant species (see Eq (13)). Unlike species abundances $x_i$, modal abundances depend only on the magnitude of ecological functions, and are therefore self-averaging. Finally, let us introduce the parameter

$$\gamma = \sigma \sqrt{C^\star}$$ (11)

that increases with the intensity of disorder and is also a self-averaging quantity.

When interactions are dominated by structure ($\sigma = 0$), the equilibrium values of species abundances coincide with the modal abundance, and are readily obtained from Eq (5) as $x_i^\star|_{\sigma=0} = \max\left(0, x_i^+\right)$. Equilibrium functional magnitudes then obey the closed set of $n_F$ equations:

$$f_\lambda^\star|_{\sigma=0} = \frac{1}{S} \sum_i \mathcal{J}_i^{(\lambda)} \max\left(0, 1 + \sum_\mu \mathcal{S}_i^{(\mu)} f_\mu^\star\right)$$ (12)

that only involve functional traits.

The distribution of species abundances in the absence of randomness can be easily visualized when the community is structured in homogeneous functional groups (Fig 1A). Every species belonging to a same group has the same traits, therefore exactly the same equilibrium abundance, which corresponds simply to the solution of the low-dimensional Lotka-Volterra equations for the functional groups. Panels A and B of Fig 3 illustrate the theoretical SAD for a community structured in four groups, where abundances of non-extinct species concentrate over just three modal values, with relative frequency equal to the fraction of species in each group. Species with negative modal abundance are driven towards extinction by their interaction with the community. Due to migration, they never go fully extinct and persist at small abundances $x_i \approx m$. Since $m \ll 1$, they do not significantly contribute to the ecology of the community, and we count them as extinct.

The addition of random interactions modifies this solution, and species will have a different equilibrium abundance even if they share the same functional traits. The effect of increasing the variance $\sigma$ is to spread the abundances around the modal abundance. Species that share the same $\mathcal{S}_j^{(\lambda)}$ will have the same $x_i^+$, consistent with the idea that they respond coherently to the function $\lambda$, despite individual heterogeneity. More precisely, in Appendix F in S1 Text we show that, when $\sigma > 0$, the equilibrium abundance of species $i$ is randomly distributed according to a truncated-Gaussian random variable,

$$x_i^\star = \max\left(0, x_i^+ + \gamma \xi_i^\star\right),$$ (13)

where $\xi_i^\star$ are independent standard Gaussian variables that reflect the diversity of traits among different species.

Remarkably, community-level variables are related by a closed set of $n_F + 1$ relations. Despite accounting for most of the dimensionality of the interaction matrix, the random component thus modifies the system in a minimal way, equivalent to adding a single macroscopic relation on top of those provided by structure. They read

$$f_\lambda^\star = \frac{\gamma}{S} \sum_i \mathcal{J}_i^{(\lambda)} \omega_1\left(\frac{x_i^+}{\gamma}\right)$$
$$1 = \frac{\sigma^2}{S} \sum_i \omega_2\left(\frac{x_i^+}{\gamma}\right)$$ (14)

PLOS Computational Biology

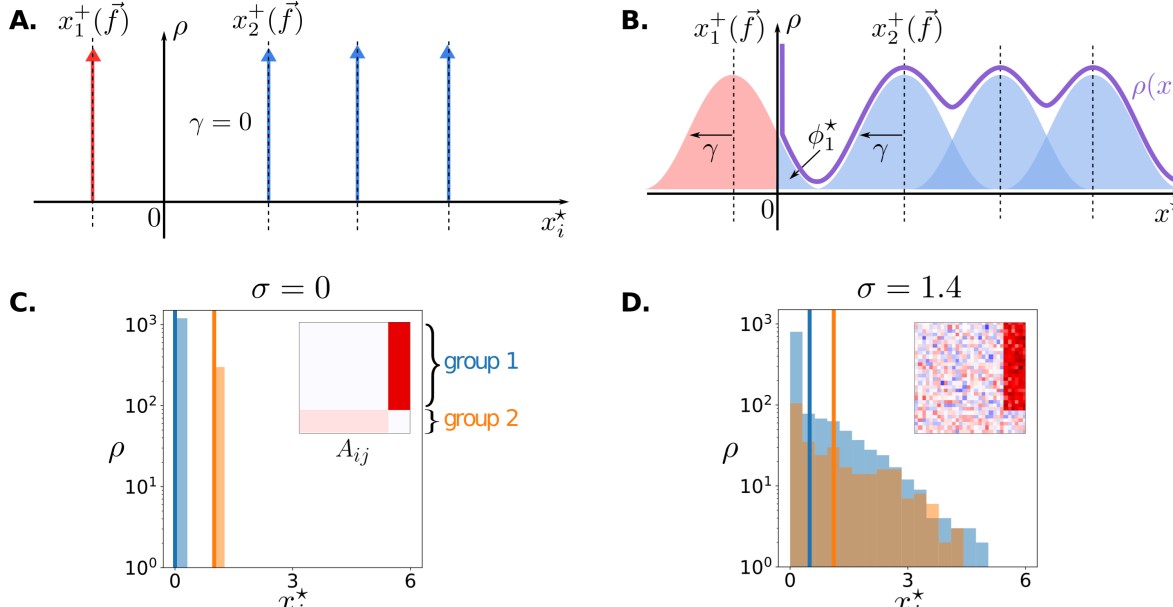

**Fig 3**. **Simple interplay in the Species Abundance Distribution.** The equilibrium SAD is a convolution of the predictions of structure and disorder. Top row: schematic illustration of the theoretical predictions of Eq (16) when the abundances are at equilibrium. Bottom row: numerical SADs for a community structured in two functional groups of different size, whose interaction matrix $A_{ij}$, plotted in the inset, is either purely structured or has a random component of intensity $\sigma$. **A.** and **C.** In the absence of randomness ($\sigma = 0$), species abundances $x_i^+$ are set by the sensitivity traits to collective functions $\vec{f}$ (impact traits being identical within groups). **B.** and **D.** When $\sigma$ increases, the distribution of species abundances within groups widens by a factor $\gamma$ around each trait-based peak $x_i^+$, but these peaks are also displaced due to changes in $\vec{f}$ (see also Fig 4). The SADs for the two groups are separately plotted in **C.** and **D.** and their mean abundances (solid vertical lines) show that a signature of the underlying structure persists even when randomness is so strong that the total SAD would appear featureless.

where the $x_i^+$ defined in Eq (10) only depend on collective variables and species traits, and we have defined the mathematical functions

$$\omega_k(w) = (2\pi)^{-1/2} \int_{-w}^{\infty} e^{-z^2/2} (w + z)^k \, dz. \tag{15}$$

i.e. the $k$th moment of a truncated Gaussian distribution centered at $w$. In particular, $\omega_0$ represents the area with positive abundance, and will be used below to define a species' survival probability.

We can now exploit the self-consistent relations Eq (14) to derive the exact distribution of the microscopic abundances. The SAD turns out to be a convolution of a 'structure-driven SAD', reflecting the deterministic biases given by the modal abundances $x_i^+$, and of a Gaussian distribution of typical width $\gamma$ (Fig 3B):

$$\rho(x) = (1 - \phi^\star) \, \delta(x) + \frac{\Theta(x)}{\gamma} \sum_i \psi \left( \frac{x - x_i^+}{\gamma} \right). \tag{16}$$

Here, $\psi(w) = \exp(-w^2/2)/\sqrt{2\pi}$ is the Probability Density Function of a standard Gaussian, $\Theta$ is the Heaviside function and

$$\phi^\star = \overline{\omega_0 \left( \frac{x_i^+}{\gamma} \right)} \tag{17}$$

is the *diversity*, that is, the total fraction of species that survive without being rescued by migration.

In Fig 3C,D we provide an example of a simulated simple community composed of two competing functional groups: each group contains a different number of species, and all species within one group share the same structural traits (interaction matrices are represented in insets). The SAD is trivial in the absence of randomness, as one group excludes the other at equilibrium (Fig 3C). When random variation is added, the excluded group is rescued, and the two peaks widen until they largely overlap (Fig 3D). Yet, the existence of an underlying structure remains detectable if the mean abundance of each group is plotted (solid vertical line). The structure-driven SAD can also feature more complex profiles that reflect the distribution of traits within the community or the functional groups. For instance, if the total abundance is the only community function and the sensitivity traits are distributed as a power-law (see appendix I.1 in S1 Text), the structure-driven SAD will have a power-law tail which progressively turns into a Gaussian shape as randomness is increased (Fig B in S1 Text). For the presence of structure to have no impact on the SAD, $\mu_{ij}$ should have particular symmetries. This happens, for instance, when the structural interactions derive from phylogenetic proximity in a balanced binary tree (see Appendix I.2 in S1 Text).

The SAD is therefore usually very sensitive to the presence of structure in the interactions. Its exact shape is dictated by the interplay of structure and randomness, and it is not expected to follow universal rules, other than those possibly imposed by universal scalings of species traits.

Even though randomness blurs the SAD, the effect of structure never vanishes in the self-consistent relations Eq (14), but the magnitude of functional variables $f_\lambda^*$ is modified by heterogeneity $\gamma$ (Fig 4). We see in these equations that increasing randomness has two effects. First, all else being equal, functional magnitudes tend to increase with $\gamma$, since abundances spread, but they are bounded only on the left by extinction (Fig 4B). Second, it causes all species to contribute more evenly to functions, due to the rescaling $x_i^+/\gamma$. This, notably, reduces the differences between species that deterministically would persist and those that would be excluded (positive or negative $x_i^+$). The probability, measured over realizations of randomness, that species $i$ persists, as per Eq (17), is different from zero and one for $\sigma > 0$. Hence, species

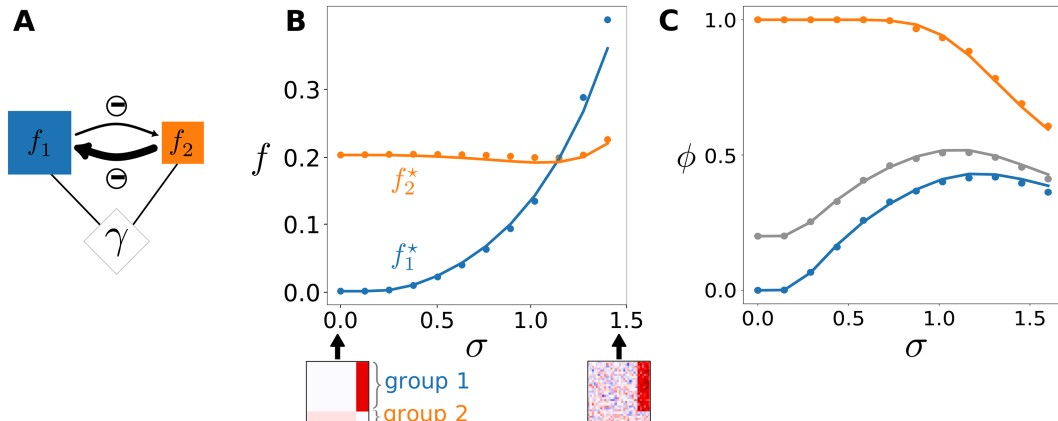

**Fig 4**. **Shift in equilibrium relations between macroscopic observables.** Even as randomness $\sigma$ increases, the collective variables $f_\lambda$ remain meaningful observables connected by deterministic functional relations, but their balance can shift due to heterogeneity $\gamma$, following Eq (14). Applied to various interaction structures, these relations translate to many ecological patterns of interest, e.g. biomass pyramids [43,49] or balance between resources [50]. **A.** Similar to Fig 3, we consider an example community structured in two competing groups: group 1 (blue) is larger but less competitive, whereas group 2 (orange) is smaller but more competitive. **B.** The functional variables (total group biomasses) change with the magnitude $\sigma$ of random interactions (dots, numerical simulations; solid lines, theoretical prediction). At $\sigma = 0$, all species in a group have the same abundance, and group 1 remains at the migration threshold. As $\sigma$ increases, heterogeneity starts to matter, and the larger group 1 can contain more "lucky" overabundant species, rescuing it from exclusion then surpassing group 2 in total abundance. **C.** This simple interplay of structure and randomness can yield non-trivial patterns. Increasing randomness initially increases total extant species diversity $\phi$ (grey: overall fraction of surviving species) by rescuing the blue group from exclusion (blue: fraction of surviving species within that group), but eventually reduces diversity in both groups and as a whole.

that would be excluded based solely on structured interactions can be "rescued" by randomness and contribute to functions, leading to non-trivial effects, even in the simple case of two functional groups illustrated in Fig 4A. In that example, group 1, initially excluded, can take advantage of randomness and drive a community-wide increase in diversity (Fig 4C). This increases the competitive pressure on group 2, and the functional dominance in the community can eventually get reversed, with the less-competitive group 1 becoming more abundant.

In summary, the addition of randomness preserves the existence of deterministic relations between macroscopic functions, but it changes the magnitude of these functions through the nonlinearities in Eq (14), notably by determining which species are extinct or not, and equalizing their contribution to functions.

### 3.3 Randomness can drive fluctuating communities to stable equilibrium coexistence

The previous analytical results rely on the existence of a stable fixed point, yet Eq (1) with structured or random interactions is known to also display out-of equilibrium dynamics, including limit cycles, neutral cycles, and chaotic behaviour [51]. In particular, it has long been theorized that adding random interactions past some critical value of $\sigma$ destabilizes communities that would otherwise be at a fixed point [1]. We show next that increasing $\sigma$ can also have an opposite, stabilizing effect on communities that would otherwise be out-of-equilibrium [52].

As an illustration, we explore the effect of introducing random variability at microscopic scale in a case where the macroscopic variables exhibit chaotic dynamics [53]. As before, the smaller scale will be called 'species' and the larger one 'groups' (whose total abundances are the associated 'functional variables'), but the argument applies for any pair of taxonomic levels, e.g. strains within species. The recent availability of data at high taxonomic resolution, as well as theoretical investigations, indeed indicate that both functional groups and species harbour a degree of genetic and trait variation, with dynamical consequences [54–57].

We consider a set of $n_F$ groups and divide each of them into $s$ species. The community is thus composed of a total $S = s \times n_F$ species that constitute the microscopic variables of our model. In the purely structured (group-scale) model, detailed in Appendix I.3 in S1 Text, a combination of strong interactions and weak immigration causes a persistent turnover between a small set of dominant groups and a large set of rare groups, thus reproducing dynamic and static patterns that are characteristic of microbial and plankton communities when observed at a high taxonomic resolution [57,58].

Fig 5 displays the asymptotic regimes of a community with fixed structural interactions and for increasing values of within-group variability. When $\sigma = 0$, all species within a group have the same chaotic trajectory as the group they belong to (Fig 5A).

When randomness $\sigma$ increases, species remain largely synchronized within groups, so that the two levels of description have the same qualitative dynamics. Their trajectories get however progressively modified and the community crosses different dynamical regimes (Fig 5B), until it reaches a stable equilibrium (Fig 5C) where numerous groups coexist even in the absence of immigration. For even larger $\sigma$, the species start to fluctuate again chaotically, but are desynchronized – so that their incoherent oscillations cancel out when abundances are averaged within groups (Fig 5D). Unlike when $\sigma = 0$, the dynamics at the species and group levels are now decoupled.

These results show that microscopic heterogeneity has different predicted effects depending on the scale of aggregation that is under scrutiny (Fig 5E). When one is interested in the microscopic scales (here, species), heterogeneity has a non-monotonic effect: it is first stabilizing and then destabilizing for strong-enough disorder (as already emphasized in many theoretical studies [1,4,22]), leading to complicated chaotic dynamics. On the other hand, the macroscopic scale (here, groups) is stabilized by intermediate heterogeneity, and fluctuates only little (and less and less if the number of species per group $s \to \infty$) even after the microscopic dynamics looses stability.

We now turn to a theoretical explanation of the dynamical regimes highlighted in Fig 5. We focus on the stability of the intermediate-randomness equilibrium, and show how the transition towards synchronized out-of-equilibrium dynamics (for small $\sigma$) and that towards microscopic chaotic dynamics (for large $\sigma$) can be related, respectively, to the random and the

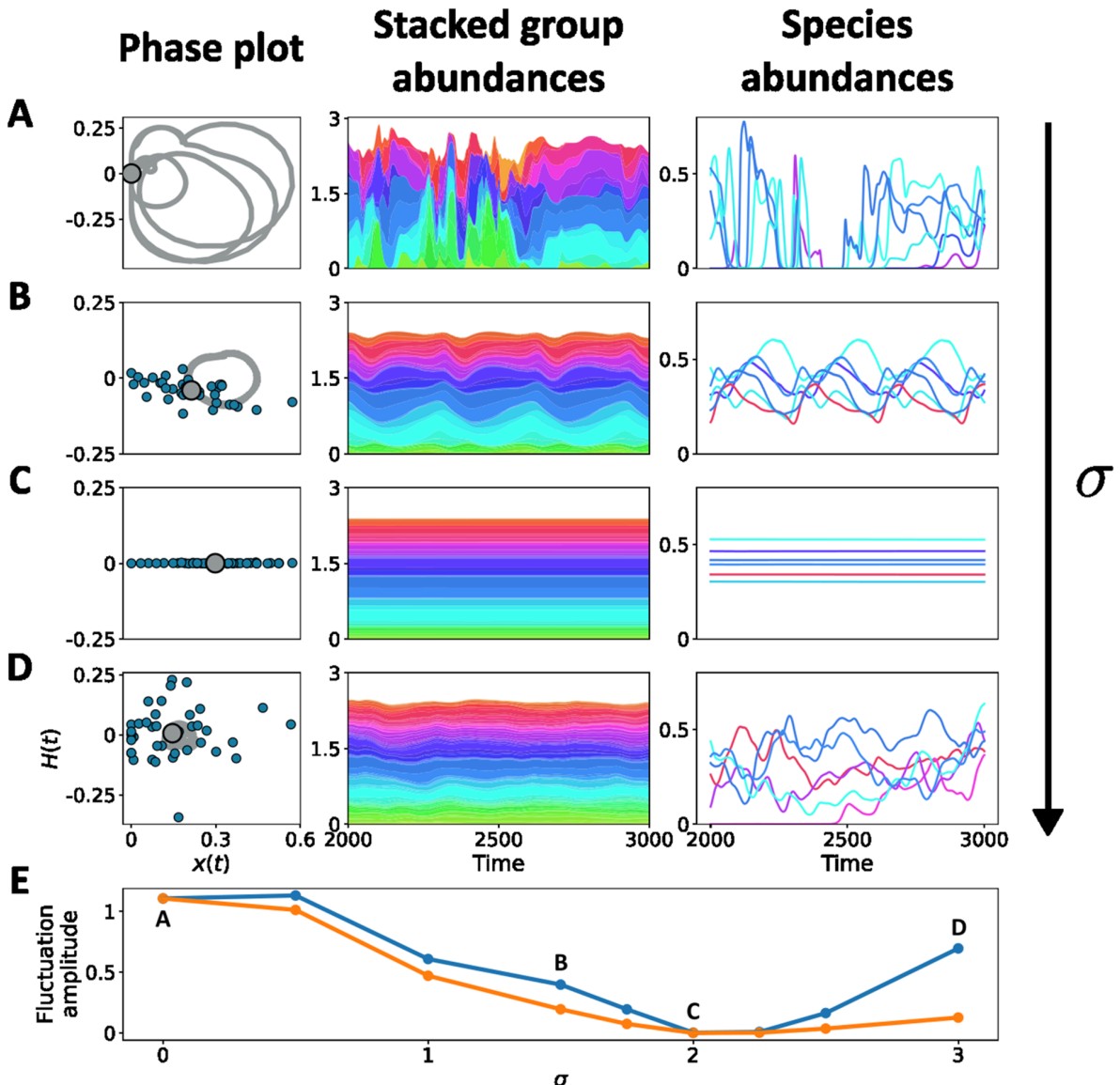

**Fig 5**. **Dynamical interplay of complex interactions at microscopic and macroscopic scales.** A matrix of group-group interactions is chosen as in Appendix I.3 in S1 Text to generate complex group dynamics. Each of $n_F = 150$ groups contains $s = 40$ species. The within-group variance of the added random interactions between species increases from top to bottom ($\sigma = 0, 1.5, 2, 3$). Left column: the trajectories of species and groups can be compared by projecting them onto a complex plane. This is achieved by plotting the Hilbert transform of the abundance against the abundance itself. Here, we consider the trajectory of the average abundance $f_1$ of the first group (solid line) and a snapshot of the abundances of individual species belonging to that group (blue dots). Middle column: stacked abundances $f_\lambda$ of groups. Right column: abundance trajectories for a couple of individual species within a few groups (of corresponding colour). **A.** When there is no disorder, species overlap with their average (gray dot), illustrating the coherence of within-group dynamics. Groups and species undergo chaotic dynamics, and within-group differences are transient effects of distributed initial conditions. **B.** As within-group mismatch in interactions is introduced, species spread out, but they remain close to their group's mean even if the qualitative dynamics changes – here it is a limit cycle. Species within groups have persistently distinct trajectories, reflecting their individual traits. **C.** For intermediate heterogeneity, oscillations are lost both at the group and at the species level, and species have different equilibria. **D.** Increasing $\sigma$ further, species become unstable through a collective transition, whereby their trajectories start to fluctuate asynchronously and chaotically. The dispersion of species belonging to a same group results in averaged-out fluctuations, which are however discernible at the group level when the community has finite size. **E.** The amplitude of fluctuations at the level of species (blue) and groups (orange) varies non-monotonically with $\sigma$: before stabilization at $\sigma \approx 2$, both are similar and decreasing, due to synchronization (as in rows **A** and **B**)); after destabilization (as in row **D**), species variance increases due microscopic chaos, but group variance remains small because the fluctuations at the species level are now asynchronous.

structured part of the interactions. The species-level instability occurring for strong randomness (Fig 5C, 5D) is akin to the *bulk-driven* collective phase transition of gLVEs with purely random interactions [4]. Both the Jacobian matrix and the reduced interaction matrix (where only non-extinct species are considered) undergo a bulk instability when

$$\phi^\star \sigma^2 - 1 = 0, \tag{18}$$

where $\phi^\star$ is the total fraction of surviving species, defined in Eq (17). While purely random systems obey the same condition, the location of the transition changes here with structure through the dependence of $\phi^\star$ on $\sigma$. The bulk-driven transition to microscopic chaos, predicted for $\sigma \gtrsim 2.25$ in the many-groups example, is confirmed by the numerical simulations.

On top of this classic collective transition, *structure-driven* transitions to equilibrium can occur as bifurcations of the coarse-grained variables. Instead of being related to the bulk of the Jacobian matrix, these bifurcations involve its outliers. Directly linked to the low-rank matrix structure $\mu_{ij}$, such eigenvalues are also modified by randomness. We show in Appendix H in S1 Text that structure-driven transitions can be located using the 'pseudo-Jacobian' matrix

$$\mathcal{J}_{ij} = -x_i^\star \left( \delta_{ij} - \mu_{ij} \right), \tag{19}$$

which explicitly depends on the structural part of the interactions, but not on randomness. Nonetheless, randomness impacts $\mathcal{J}$ indirectly through the equilibrium abundances $x_i^\star$, see Eq (13). Thus, structure-driven transitions are signaled by eigenvalues of Eq (19) having a vanishing real part. To illustrate this, we report the spectrum of $\mathcal{J}$ for the system of Fig 5 in Fig A in S1 Text.

Whereas the large-$S$ collective transition Eq (18) involves an extensive number of bulk eigenvalues becoming unstable at once, here the increase of microscopic heterogeneity can cause a single pair of conjugate outliers to cross the imaginary axis through a Hopf bifurcation. The mathematical underpinning of how microscopic randomness generically suppresses collective fluctuations is addressed elsewhere [52].

Here, we used this example to stress the fact that microscopic randomness and macroscopic structure may appear to impact fixed point stability through independent pathways (corresponding to bulk and outlier eigenvalues of the interaction matrix, respectively), yet microscopic randomness can actually impact and even stabilize complex structure-driven dynamics through its simple effect on equilibrium abundances, as shown in Fig 3.

## 4 Discussion

Models of ecological dynamics with random species interactions are increasingly explored as a paradigm for simple and parsimonious explanations of species abundances and dynamics [5]. Yet, we do not know to which extent their predictions are robust to the presence of structure in species interactions, such as the existence of functional groups or other macroscopic features, evidenced in many empirical interaction networks and also exploited to explain community-level patterns [8]. In particular, it is important to ascertain whether including both randomness and structure in species interactions may still allow for simple predictions, or whether we should expect the increased complexity to severely hamper generalization.

As a first step toward understanding the possible interplay of randomness and structure, we have considered a model for community dynamics where we know that parsimonious predictions are possible both when interactions are purely structured and when they are purely random (Fig 2). When structure is the only determinant, simplicity arises from the existence of a limited number of macroscopic observables, which represent broad ecological functions of the community (e.g. the abundance of various groups such as predators and prey, or the availability of key resources) and control how every species affects and responds to others. On the other hand, species-rich communities with purely random interactions display collective regimes whose statistical regularities can be captured by just a few macroscopic observables. In

general, we can expect simple structuring processes and complex random-like processes to play out in parallel in species interactions. Modeling this as a simple additive superposition, we derived a single effective description of the community dynamics that integrates these two limits.

We investigated several aspects of the interplay between randomness and structure, associated to different mathematical properties of the dynamical model: equilibrium patterns of (i) species abundances (SADs) and (ii) macroscopic variables, and (iii) transitions to out-of-equilibrium dynamical regimes (Fig 2). While this interplay can lead to a variety of outcomes depending on the specific structures considered, we focused here on broadly relevant patterns. In particular, several non-trivial results could be understood as consequences of two key mathematical phenomena. First, increasing randomness in species interactions causes species abundances to spread around their values predicted by structure. Second, randomness creates a bulk of eigenvalues in the spectrum of the interaction and Jacobian matrices, while the existence of a small number of structuring collective functions creates outlier eigenvalues - with the two parts of the spectrum producing different pathways to instability. These generic phenomena are robust to model details, and come together to explain a variety of results.

At the microscopic level, predictions from purely random models for Species Abundance Distributions are readily altered when structure is present (Fig 3). This is consistent with prior findings that a small, low-rank perturbation of the interaction matrix is sufficient to modify or fully determine species abundances [6,59]. In particular, the shape of the SAD reflects the distribution of traits that make up the structural matrix. On the other hand, many different structural matrices can produce the same SAD, which is thus not a strong fingerprint of underlying processes.

At the macroscopic level, remarkably, the equilibrium state can always be described in terms of a small number of community-wide variables (Fig 4). A set of relations, Eq (14), connects the community-level ecological functions that structure the interactions (associated with functional groups, public goods, etc.) and a single additional degree of freedom that represents randomness-induced heterogeneity. These relations underlie many ecological patterns, e.g. how the biomass of a predator may depend on that of prey [49]. We found here that these relations exist for all levels of heterogeneity, but their equilibrium is displaced as it increases in magnitude. As a consequence, signatures of ecological structure may be detected despite random variation, if one knows how to measure the relevant functions.

When the community dynamics is not at equilibrium (Fig 5), random interactions qualitatively modify the community dynamics in essentially two, independent ways, as revealed by analyzing the stability of equilibria. First, purely random models have shown a phase transition to a regime where species abundances fluctuate incoherently, thus averaging out in aggregated observables such as mean abundance. This is associated to the destabilization of the bulk of eigenvalues, that is highly robust to the addition of macroscopic structure (correlations can delay or hasten this transition quantitatively [60] but do not modify it qualitatively). Second, in virtue of its effects on species abundances, randomness can modify the community dynamics induced by structure by displacing the outlier eigenvalues associated to the collective modes. Abundances thus play a nontrivial role in determining the stability of the community attractors.

Tying these findings together, we can heuristically posit a hierarchy of robustness among the qualitative predictions of purely random models. Species Abundance Distributions, on the one hand, are very fragile to the presence of structure. On the other, the emergence of asynchronous species-level fluctuations for large microscopic heterogeneity is more robust. Moreover, addition of randomness to out-of-equilibrium structured communities can produce another, generic transition to stability for high-enough heterogeneity (Fig 5) [52].

Our description shows that the effect of randomness is channeled through a very simple mechanism: the spread in the abundances caused by randomness. In the SAD, this smooths out the structural patterns, hence reducing the importance of finely-varying structural traits (Fig 3). It allows some species to become dominant and magnify any ecological function they contribute to (typically favoring functions that receive contributions from more species, Fig 4). It induces species-level exclusion and a bias toward overall less competitive interactions among survivors [61], but it also moderates the impact of large-scale (e.g. group-level) exclusion. Finally, species abundances factor strongly into dynamical time scales, thus

abundant and rare species play different roles in community stability [62], and a randomness-induced spread in abundances and timescales can deeply alter the dynamics. This suggests a possible analogy between randomness-induced stabilization and other phenomena in intrinsically out-of-equilibrium systems, such as the phenomenon of amplitude death in populations of globally coupled oscillators, where time scale mismatch also induces a synchronous gradual unfolding of coherent dynamical regimes [63].

Simple structure and pure randomness have emerged as contrasting paradigms, both promising understanding and prediction of broad phenomenological features from limited information. The increasingly common deployment of techniques to characterize communities on multiple scales (e.g. genomic and metabolic) calls for a synthesis of these two approaches. Our theoretical results establish baseline intuitions on how to combine these two approaches to explain a variety of community patterns. We expect these intuitions to remain broadly relevant even in more complex settings. Still, it is worth discussing our simplifying assumptions. Some of them are easily lifted, e.g. identity in species growth rates $r$ and carrying capacities $K$ can be relaxed. Others pose instead greater theoretical challenges, such as when the disordered matrix either has correlations, or is perturbed by a sparse or localized structured component [21]. Our choice to disregard correlations reflects the idea that important symmetries are already represented in the structural matrix, but correlations between $A_{ij}$ and $A_{ji}$ are often treated as a meaningful parameter [22], and progress might also be made for more complicated correlations, such as those addressed in Random Matrix Theory [64] and with gLVEs [65].

Finally, this work has focused on theoretical principles, but provides some hints on possible signatures of the presence of hidden interactions in data. For instance, communities structured in functional groups may be compared across environments that preserve group-level interactions. This may teach us the role of hidden variability and whether we should trust qualitative and quantitative conclusions of models drastically less complex than natural systems.

## Supporting information

**S1 Text. S1 contains details on the numerical procedures used to simulate dynamics and solve the equations, as well as details of the computations leading to the results of the main text.** S1 also contains some examples of results for particular interaction structures.
(PDF)

## Acknowledgments

The authors are very grateful to Jean-François Arnoldi for his participation in the initial stages of the project, and Emil Mallmin for fruitful discussions and for sharing his simulation code.

## Author contributions

**Conceptualization:** Juan Giral Martínez, Matthieu Barbier, Silvia De Monte.

**Formal analysis:** Juan Giral Martínez.

**Investigation:** Juan Giral Martínez, Matthieu Barbier, Silvia De Monte.

**Supervision:** Juan Giral Martínez, Matthieu Barbier, Silvia De Monte.

**Writing – original draft:** Juan Giral Martínez, Matthieu Barbier, Silvia De Monte.

**Writing – review & editing:** Juan Giral Martínez, Matthieu Barbier, Silvia De Monte.

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
