## [Decision Letter · Decision Letter 0]

2 Jun 2025

PCOMPBIOL-D-25-00663

Structured and random interactions interplay in complex ecosystems dynamics

PLOS Computational Biology

Dear Dr. Giral Martínez,

Thank you for submitting your manuscript to PLOS Computational Biology. After careful consideration, we feel that it has merit but does not fully meet PLOS Computational Biology's publication criteria as it currently stands. Therefore, we invite you to submit a revised version of the manuscript that addresses the points raised during the review process.

All three reviewers agree that your manuscript has the potential to make a significant contribution as it bridges classic disordered-interaction models to function-based ecological structure, and it showcases how DMFT can be extended to this hybrid setting. At the same time, they all raise concerns about clarity of presentation. In addition, both Reviewer 1 and Reviewer 3 point out that the manuscript needs to be better situated within the existing literature to clarify its novel contributions.

I find myself in agreement that the presentation, including the figures as noted by Reviewer 2, undersell your results. For example, outside of the plankton model, it is never entirely clear what the coarse-grained quantities are—e.g. if total biomass of functional groups, then what defines a functional group? I also found the discussion to be quite technical and likely to be understood and appreciated only by a very specialized audience. In your revision, you should translate your mathematical results and their significance to a broader audience, more clearly identifying which aspects of the community are highly sensitive to hidden variation and which appear broadly robust, and spell out what this means for noise-driven changes in stability and dynamics.

Please submit your revised manuscript within 60 days Aug 02 2025 11:59PM. If you will need more time than this to complete your revisions, please reply to this message or contact the journal office at ploscompbiol@plos.org. Please include the following items when submitting your revised manuscript:

We look forward to receiving your revised manuscript.

Kind regards,

Rafael D'Andrea, Ph.D.

Academic Editor

PLOS Computational Biology

Zhaolei Zhang

Section Editor

PLOS Computational Biology

**Additional Editor Comments :**

L170. What are the coarse-grained variables?

P8 Footnote: what are the u’s and v’s?

Figure 2: If I understand this figure correctly, I believe the black blocks should be on entries (1,3), (2,1), (3,2) rather than on (1,2), (2,3), (3,1). Also, the figure needs to better explain that the dark segments represent sensibility (sensitivity?) and impact of species *i* with respect to group lambda, which is not at all clear.

L207. These effect traits are never explained. Are they the same as calligraphic S?

L234. It is not at all clear from Eq. 15 that this represents the survival probability.

L241. This needs a more detailed explanation.

Fig. 4 caption: (d) proportion, not number, of surviving species.

L243. Wouldn’t the observed diversity in your simulations depend on the abundance cutoff used to determine which species survive?   

**Journal Requirements:**

3) We notice that your supplementary information is included in the manuscript file. Please remove them and upload them with the file type 'Supporting Information'. Please ensure that each Supporting Information file has a legend listed in the manuscript after the references list.

4) Please amend your detailed Financial Disclosure statement. This is published with the article. It must therefore be completed in full sentences and contain the exact wording you wish to be published.

2) If any authors received a salary from any of your funders, please state which authors and which funders.

5) Please ensure that the funders and grant numbers match between the Financial Disclosure field and the Funding Information tab in your submission form. Note that the funders must be provided in the same order in both places as well. Currently, "Frontiers in Research and Education graduate program" is missing from the Funding Information tab.

**Reviewers' comments:**

Reviewer's Responses to Questions

**Comments to the Authors:**

**Please note that two reviews are uploaded as attachments.**

Reviewer #1: The review is uploaded as an attachment.

Reviewer #2: The manuscript by Giral Martinez et al applies advanced analytical techniques to address an important and timely question: to what extent do the predictions of random models (a staple of a large subfield of theoretical ecology) apply in structured communities? To address this, the authors introduce a clever model that combines strcutured and disordered interactions in a way that is both plausible and combines the strengths of the two dominant modeling frameworks (Lotka-Volterra, which is in some sense "very general" but purely compositional, and more "functional" frameworks like resource competition or functional guilds). They show that a powerful analytical technique (DMFT) can be productively applied in their model, and use it to draw some conclusions about the dynamics. The manuscript is an impressive contribution, and with some changes could be quite impactful. However, I do think a few changes to presentation may be required for this work to actually attain this impact.

My main concern is that the work, as written, will fail to reach the target audience. The model and the approach proposed here are striving to do something big and impactful. Today, there is a big tension: powerful stat physics methods can calculate so much, but in the context of "silly" random models that seem very far from anything real. This manuscript is an important step towards resolving this tension. The authors have made some commendable decisions, e.g. explaining the idea of DMFT in reasonably accessible way, by packaging away much of the technical detail into the supplement, and by making this supplement quite detailed and reasonably pedagogical. I applaud this effort. However, in my mind, the "punchline" figures in the main text entirely fail to communicate what the manuscript is achieving - no biologist will find them relatable or impressive (and physicists would arguably struggle with this as well…).

In summary, I believe this work is potentially very impressive, but is let down by serious issues with presentation.

Major issues:

(1) This is by far the biggest. To me, Figs. 4-6 aren't working. It is not my role to tell the authors what the figures should rather be - they are the best judges of what constitutes the most impressive aspects of their results. However, the readership of PLOS Comp Biology is generally not made of statistical physicists, and even physicists skimming the figures wouldn't know why these results are supposed to be exciting. So I urge the authors to recruit a biologist friend and ask them for feedback on these figures, especially 4 and 5. I am quite certain that they massively undersell the actual results.

Here are some things that might help.

(1a) Consider replacing figure titles by a statement of what one is meant to conclude, not what is plotted.

Let's take Fig. 6. Current title: "Influence of strain variability on species and strain dynamics". In reality, the figure shows that randomness can stabilize oscillating communities, but in a very interesting non-monotonic way, with a disorder-induced transition. This is fascinating, and would be completely lost on the reader based on the title. Thankfully, here the figure is visual enough that this may still come across. But Fig. 4 & 5 are not so fortunate:

• "Fig 4: Species abundance distributions and community-level observables at equilibrium" - what about these SADs should the reader find interesting? That they are well predicted by the model? That the structured interactions changes them dramatically? Doesn't change them at all? That there are multiple regimes? What is the biologically relevant statement? Etc. etc. Fig. 4a and 4b almost seem like they are illustrating a minor technical point. Is this the key result that deserves being showcased in the prime real estate of the punchline figure Fig. 4? What feature of Fig. 4cd are we meant to be impressed by? That these curves increase, decrease? That the lines pass through the dots? that panels c and d are different?

• "Fig 5: Species abundance distributions for communities with a power-law distributed structural trait." Why are we interested in a power-law distributed structural trait? This title makes the figure seem like some minor technical detail.

Some of these points are explained much better in the text. BUT people look at the figures to decide whether to read the text. Poorly designed figures will obliterate the impact of the paper. And furthermore, crystallizing out the impact that would speak to biologists would improve the text as well (even for physics-trained readers).

(1b) The premise of the paper is to investigate how predictions or behaviors change when one tuens the realtive impact of structure vs random interactions. This, I believe, is the entire point & strength of the model - to provide a knob one could turn to interpolate between "fully random" and "highly structured". But then, the punchline plots has to be "something against sigma": the two extremes that are well understood, the middle that is new, exciting, and interesting. In the current exposition, the only such plot is the very last panel of the very last figure.

(2) Continuing with the presentation issues: I found Figs. 1 & 2 hard to parse.

• In Fig. 1 I'd recommend making the arrowheads larger to be more visible.

• Legend refers to lambda, but there is no lambda in the figure. Is lambda labeling the box (green/blue)?

• In both figures, I think readability could be greatly enhanced by a strategic placement of labels ("species; functions; effect traits, sensitivity traits") that would make notations understandable without constantly referring to the figure legend. In Fig. 1, it might be helpful to explicitly label the two sides along the lines of "interactions structured by functional groups" and "resource-mediated interactions".

• In Fig. 1a, it looks like the two black arrows are jointly labeled by S^(1)_j. Perhaps labeling the other arrow as S^(1)_i would make this figure easier to understand. (Fig. 1b is better in this sense, with all three arrows labeled correctly).

• A separate legend for i and j being two *green* shapes in confusing. The existence of same shapes in both green and blue box is also confusing. Is this intentional? Are the blue and green dots the same species (that can contribute to two groups), or are they different species?

• To non-programmers, starting the numbering from zero is confusing. It seems to imply that 0 is somehow special. Besides, Fig. 1 numbers the two groups as 1,2 and not 0,1.

• In Fig. 2 I can understand the "plus" between (a,b) and (c,d), but the plus between species is confusing as it suggests the vectors are added, which they are not.

(3) This is more of a "medium" issue, and has to do with contextualization of the study.

Our goal here is to investigate which predictions of fully random models are altered by structure in species

interactions. Among possible predictions, only equilibrium stability has received extensive attention along

this line so far.

I agree, but I'm not sure this passage is entirely fair. The value of the question is recognized rather broadly. It's true that within gLV-based work, an outsized fraction has focused on equilibrium stability, but gLV simply doesn't offer very many options in terms of relevant observables, so diversity and stability reign supreme. But investigating the extent to which random-model predictions survive relevant levels of structure (including how this interacts with specifically the ability to coarse-grain ecosystems) is already recognized to be broader. Some theorists interested in similar questions choose to use resource competition models (Ben Good, Mikhail Tikhonov) -- precisely because that framework allows accessing other (more "functional") properties. There's also the question of the extent to which large-scale model predictions depend on other interaction-structuring "details", such as metabolism (Avi Flamholz, Daniel Segre, Terry Hwa) or resource preference hierarchies (Sergei Maslov, Akshit Goyal).

Obviously, there's no need for the authors to review *all* that work here, but mentioning some of this would actually help underscore the cleverness of the authors' approach as it actually marries gLV with elements of those "functional" models. It would also make their work accessible & appealing to a broader readership. As written, I was originally slightly uncomfortable with the authors' framing of their work as being in contrast to "limited" previous discussions ("previous work was too specific, and this calls for the investigation of whether there exist any general principles ruling the interplay of randomness and structure"), because their introduction only referenced a small subset of the relevant literature.

This is not at all a criticism of the study (in fact, I think this entire broader literature would benefit from the authors' contribution here), merely a suggestion to perhaps slightly expand the framing/contextualization in the introduction:

• perhaps replacing "only equilibrium stability has received attention" by "equilibrium stability received particular attention, but…"

• And perhaps by adding a few more sentences and references out of the names above, plus Allesina (e.g. Allesina et al 2015 https://doi.org/10.1038/ncomms8842), O'Dwyer; maybe Crocker et al (biorxiv 635766)? -- whoever the authors feel is most relevant, to help orient the reader in the broader context (both within and beyond gLV).

Minor issues:

Line 57: citing reference 9 here is consistent with the purported conclusion of that paper, but the authors may wish to consult https://doi.org/10.1101/2024.03.26.586891 which casts doubt whether that example does in fact belong in this list.

Would "sensitivity" be a better term in this context than "sensibility"?

• Sensitivity: the capacity of an organism or organ to sense and respond to stimulation

• Sensibility: the capacity to perceive or feel, particularly in relation to emotions, aesthetic appreciation, or moral values; sensitivity.

Lines 170-171: I'm not sure I understood this sentence. We are meant to conclude that in this regime, everything is straightforward, but what are these ODEs that collective quantities satisfy? Eq. (5) expresses species dynamics in terms of collective quantities, but the latter are in turn composed of species dynamics. Am I missing something obvious?

Reviewer #3: See attached file

**Have the authors made all data and (if applicable) computational code underlying the findings in their manuscript fully available?**

Reviewer #1: None

Reviewer #2: Yes

Reviewer #3: Yes

PLOS authors have the option to publish the peer review history of their article (what does this mean?). If published, this will include your full peer review and any attached files.

Reviewer #1: No

Reviewer #2: No

Reviewer #3: No

**Figure resubmission:**
---

## [Decision Letter · Decision Letter 1]

2 Oct 2025

PCOMPBIOL-D-25-00663R1

Interplay of structured and random interactions in complex ecosystems dynamics

PLOS Computational Biology

Dear Dr. Martinez,

Thank you for submitting your manuscript to PLOS Computational Biology. After careful consideration, we feel that it has merit but does not fully meet PLOS Computational Biology's publication criteria as it currently stands. Therefore, we invite you to submit a revised version of the manuscript that addresses the points raised during the review process.

Please submit your revised manuscript within 30 days Dec 02 2025 11:59PM. If you will need more time than this to complete your revisions, please reply to this message or contact the journal office at ploscompbiol@plos.org. Please include the following items when submitting your revised manuscript:

We look forward to receiving your revised manuscript.

Kind regards,

Rafael D'Andrea, Ph.D.

Academic Editor

PLOS Computational Biology

Zhaolei Zhang

Section Editor

PLOS Computational Biology

**Additional Editor Comments :**

I concur with all three reviewers that this is a substantially improved presentation of your work. While Reviewers 1 and 3 are satisfied with the current version, Reviewer 2 and I have some minor comments, particularly regarding the figures, which may help improve the manuscript further.  

L23. This sentence is ambiguous. Are the patterns less robust to breaking equivalence than they are to the qualitative dynamics, or are the patterns less robust to breaking equivalence than the qualitative dynamics are?

L141. “Impact traits” should be “sensitivity traits”

L203. cancel out (no hyphen)

Fig 1. The matrix examples remain unclear. For instance, which aspect of matrix B tells us that there are two public goods here (water and pH)? In matrix C, which square size corresponds to which taxonomic unit (species, genus, family)? Matrix A should be especially clear since functional groups are the focal structure in the study. It can improve if the reader can connect the colors of the squares, which represent interactions between the functional groups, to the width of the arrows in the diagram. I suggest making all arrow widths visibly different and using a color gradient for the squares that makes their magnitude clear without needing a legend (which is understandably omitted).

Fig 2. Neither the caption nor axis labels fully explain what is being plotted in the equilibrium plots. It looks like those are species abundance distributions, with abundance on the x-axis and number of species on the y-axis. But this should be explicitly mentioned (for example, <img height="17" src="data:image/png;base64,iVBORw0KGgoAAAANSUhEUgAAAAgAAAARCAMAAADXCB3qAAAAAXNSR0IArs4c6QAAAEhQTFRFAAAAQEBAQECMQGiMQIzIaGiMaKvljEBAjGhAjMjljMj/q2hoq+X/yIxAyIxoyP//yOX/5ato5ciM/8iM/+Wr/+XI///I///lTWwPoQAAAAF0Uk5TAEDm2GYAAAAJcEhZcwAADsQAAA7EAZUrDhsAAAAZdEVYdFNvZnR3YXJlAE1pY3Jvc29mdCBPZmZpY2V/7TVxAAAAQklEQVQYV2NgoBSIcjAy8wANEWPnZOBjATIEgQQIM/ByATEbA4M4Nw+DEBM/A4MIK1CtAFBGGCQNAoJAJWAAVIIKAFU4AYkQhG7iAAAAAElFTkSuQmCC" width="8" /> should be defined, or better still, replaced with words).

In the caption to Fig 3 and elsewhere in the text, it is repeatedly stated that when the magnitude of random interactions is set to zero, the blue group (group 1) is extinct. However, this model includes immigration. To avoid confusion, the language should be changed to something along the lines of “group 1 subsists at background levels, rescued from extinction by immigration.”

L280 “past some critical value of <img height="17" src="data:image/png;base64,iVBORw0KGgoAAAANSUhEUgAAAAgAAAARCAMAAADXCB3qAAAAAXNSR0IArs4c6QAAAEhQTFRFAAAAQEBAQEBoQGirQIyrQIzIaECMaGhAaGiraKvljEBAjKvljMj/q4xoyIxoyOX/yP//5ato5ciM/8iM/+Wr/+XI5f/////lLgSuowAAAAF0Uk5TAEDm2GYAAAAJcEhZcwAADsQAAA7EAZUrDhsAAAAZdEVYdFNvZnR3YXJlAE1pY3Jvc29mdCBPZmZpY2V/7TVxAAAAOUlEQVQYV2NgoBQIsTMyMvIwMPBx8AtzAg0TYRNgEOEQY2AQZmVgEGQGMVjERLlAUuK8jEzcRNoHAGIfAY9peb9IAAAAAElFTkSuQmCC" width="8" />” is a restrictive modifier, so it should not be set off with commas.

Fig. 5 caption: remove duplicated word “group” before “contains” on the third line.

L388: typo “A**s** a consequence”

On a general note, I disagree with the implied interpretation of the <img height="17" src="data:image/png;base64,iVBORw0KGgoAAAANSUhEUgAAAAgAAAARCAMAAADXCB3qAAAAAXNSR0IArs4c6QAAAEhQTFRFAAAAQEBAQEBoQGirQIyrQIzIaECMaGhAaGiraKvljEBAjKvljMj/q4xoyIxoyOX/yP//5ato5ciM/8iM/+Wr/+XI5f/////lLgSuowAAAAF0Uk5TAEDm2GYAAAAJcEhZcwAADsQAAA7EAZUrDhsAAAAZdEVYdFNvZnR3YXJlAE1pY3Jvc29mdCBPZmZpY2V/7TVxAAAAOUlEQVQYV2NgoBQIsTMyMvIwMPBx8AtzAg0TYRNgEOEQY2AQZmVgEGQGMVjERLlAUuK8jEzcRNoHAGIfAY9peb9IAAAAAElFTkSuQmCC" width="8" /> term in Eq. (4) as standing in contrast with or independent from functional traits (e.g. line 409). Rather, it arguably stems from contributions to the interaction matrix from unobserved or unmodelled traits and/or functions. It is quite distinct from random demographic or environmental fluctuations, which can affect changes in dynamics and coexistence outcomes *apart from* the influence of species interactions. This could be easily addressed by a note early on clarifying that the random component of the interaction matrix reflects latent ecological structure, not extrinsic noise.

**Journal Requirements:**

1) We note that your Supplementary Figures files are duplicated on your submission as they are uploaded separately and included in the supporting information file. Please remove any unnecessary or old files from your revision, and make sure that only those relevant to the current version of the manuscript are included.

Note: Supplementary figures should be uploaded with the file type 'Supporting Information' rather than 'Figure'.  Please ensure that each Supporting Information file has a legend listed in the manuscript after the references list.

2) We have noticed that you have uploaded Supporting Information files, but you have not included a list of legends. Please add a full list of legends for your Supporting Information files after the references list.

**Reviewers' comments:**

Reviewer's Responses to Questions

Reviewer #1: The authors have increased the quality of the presentation of their results. I believe that the current version of the manuscript is suitable for publication.

Reviewer #2: I thank the authors for making the extensive revisions. I find the clarity of the manuscript to be greatly improved; which I hope will help the manuscript achieve the impact it deserves. I also appreciated the reorganizing of some material in to the SI vs main text. Unfortunately, the scale of changes may have created a couple rough edges that could use some smoothing (e.g. a notation not defined until later, or an abrupt / seemingly unmotivated narrative transition.). I mention these minor issues below.

My one larger new comment is about Figure 5. I find the reorganized version to be rather unsatisfying. Previously, there was a clear punchline: a non-monotonic behavior, and the illustrative panels were followed by a summary panel delivering this message. Based on the response to the referees, the authors now feel uncomfortable calling out this "nonmonotonicity", since (as they explain in the response to reviewer 3) "it is not the same chaos", but actually two separate effects.

To be honest, I would see no harm in calling this "nonmonotonic" -- the variance goes down and then goes back up. The authors understand this well, to the point that they can explain that the origin of the chaos in the two cases is different. This seems like a point that should make the narrative stronger, not weaker: "look, the variance behaves non-monotonically, and we understand exactly where this is coming from!"

But the authors' chosen solution took all the punch out and made this figure completely "toothless". Although the summary panel has been dropped, the rest of the figure presentation remains extremely suggestive of non-monotonicity (the arrow of sigma connecting the panels; chaos, then no chaos, the chaos again…). But now this suggestiveness never goes anywhere. Merely showing a few panels with *examples* of dynamics with no systematic quantification seems rather weak. A reader skimming the figures (before reading the text) is not presented with any coherent summary or punchline for this figure, meaning they are invited to draw some conclusion themselves from the examples shown (never a good idea). But worse, the most natural conclusion is conspicuously -- and (for a reader who did NOT read the response to reviewers) inexplicably! -- avoided, and the text appears to dance around this point (e.g. the word "non-monotonic" has been scrubbed from both legend and text). To me it seems that the desire to be extra precise turned an interesting point into something much weaker and confusing.

Minor points:

• Figure 1B, bottom - while other bottom-row illustrations are self-explanatory, for this one it might be helpful to clarify in the legend that this illustration is meant to illustrate a low-rank structure (induced by this competition/influence of public goods).

• Figure 2 legend never defines x*. In fact, this figure is first referenced in line 197, but x* isn't defined until line 237 (which is actually full two pages later -- please note that line numbering in LaTeX is incompatible with some equation environments, and half of your lines aren't numbered…)

• In Figure 4, the graph of how f1* and f2* interact is not very clear (and has no panel label). Also, the legend should specify what lines vs dots mean.

• Figure 4B - I'd suggest explicitly reminding the reader what quantity is denoted "phi" denotes (would be especially appreciated by reader scanning figures before reading the paper text).

• Figure 4C: The claim of non-monotonicity in "both groups" is not exactly spectacular. For both blue and gray lines it hinges on a single (last) point. Would this be supported more strongly if you included a few more points on the large-sigma end?

• Unnumbered line following line 323:

We now turn to a theoretical explanation of the two possible ways species can transition out of the intermediate-randomness equilibrium.

I found this transition a little jarring; the flow could be improved.

Reviewer #3: I sincerely thank the authors for their efforts in revising the manuscript, which, in my opinion, is now suitable for publication.

**Have the authors made all data and (if applicable) computational code underlying the findings in their manuscript fully available?**

Reviewer #1: None

Reviewer #2: Yes

Reviewer #3: Yes

PLOS authors have the option to publish the peer review history of their article (what does this mean?). If published, this will include your full peer review and any attached files.

Reviewer #1: No

Reviewer #2: No

Reviewer #3: No

**Figure resubmission:**
---

## [Editor Report · Decision Letter 2]

25 Nov 2025

Dear Dr. Martínez,

We are pleased to inform you that your manuscript 'Interplay of structured and random interactions in complex ecosystems dynamics' has been provisionally accepted for publication in PLOS Computational Biology.

Best regards,

Rafael D'Andrea, Ph.D.

Academic Editor

PLOS Computational Biology

Zhaolei Zhang

Section Editor

PLOS Computational Biology

---

## [Editor Report · Acceptance letter]

PCOMPBIOL-D-25-00663R2

Interplay of structured and random interactions in complex ecosystems dynamics

Dear Dr Giral Martínez,

I am pleased to inform you that your manuscript has been formally accepted for publication in PLOS Computational Biology. Your manuscript is now with our production department and you will be notified of the publication date in due course.

With kind regards,

Anita Estes
